# Target Scattering Feature Extraction Based on Parametric Model Using Multi-Aspect SAR Data

Xiaoyang Yue [1,2,3] , Fei Teng [1,2]*, Yun Lin [4] and Wen Hong [1,2]

1 Aerospace Information Research Institute, Chinese Academy of Sciences, Beijing 100194, China
2 Key Laboratory of Technology, Geo-Spatial Information Processing and Application Systems, Aerospace Information Research Institute, Chinese Academy of Sciences, Beijing 100190, China
3 School of Electronic, Electrical and Communication Engineering, University of Chinese Academy of Sciences, Beijing 100049, China
4 School of Electronic Information Engineering, North China University of Technology, Beijing 100144, China
* Correspondence: tengfei16@mails.ucas.ac.cn; Tel.: +86-188-1051-0903

**Abstract:** The multi-aspect SAR observation can obtain the backscattering information of the illuminated scene target. There are targets with different structures in the scene, and their backscattering responses are also different. Using backscattering amplitude information to analyze the differences between targets is a conventional method. For point target types, one-dimensional backscattering curves can be used to analyze scattering characteristics, but it is difficult to analyze the overall structure of the target. Therefore, it is necessary to perform statistical analysis on the backscattering information combining with the multi-aspect target area, and establish parameters to model the target area. In this paper, the algorithm uses the $G^0$ distribution based on expectation maximization (EM) to fit the target area of the SAR scene. For different target types in the scene, the $\beta$ and $\sigma$ parameters obtained by the model combined with the backscattering amplitude information are used to perform the target. The results show that full-target in multi-aspect SAR image can be differentiated by two parameters. The scattering of partial-target slices can be characterized using two parameters (amplitude difference from surrounding points, scattering energy). The parametric model quantitatively characterizes the scattering feature of the target level, and the parameters changing corresponds to the change of the target image feature. C-band circular SAR data is used to validate our method. The experimental results give the parameter representation with sampling window based on the analysis of the target scattering, and give parameter estimates to characterize the partial target scattering.

**Keywords:** multi-aspect; backscattering; distribution; parametric; target

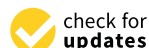



## 1. Introduction

Multi-aspect synthetic aperture radar obtains echo data by means of the azimuth changing of the observation platform [1–3]. Due to the rich target information in the scene: the difference of target structure, the change of target state, the fluctuation of scene terrain, etc., the scene information can be analyzed by means of multi-aspect SAR. Shen et al. established a model for moving targets using single-channel circular track SAR [4]. Feng et al. extracted terrain elevation information by means of multi-aspect observation of terrain changes [5]. In addition, there are multi-aspect polarization classification and target characteristics analysis, etc. [6,7].

The synthetic aperture radar for scene can obtain the backscattering information of the target in the corresponding scene, that is, the response of the target to the radar illumination at different observation angles [8–10]. Traditional SAR modes such as strip SAR have limited azimuth observation, so the backscattering information of the target is limited. Due to the addition of azimuth observation information, multi-aspect SAR can observe the target scene from more angles and obtain more target backscattering information [11–13].

Li et al. proposed an anisotropic scattering detection method based on the multiangular fully polarimetric signatures [7]. Feng et al. combined multi-aspect SAR with vehicle geometry for vehicle 3D elevation extraction [14]. Ai et al. proposed a novel CNN model based on multi-channel size feature fusion for SAR target classification, improving the analysis ability of SAR target characteristics [15]. At the same time, a description method for ship's feature extraction is established using the multi-scale rotation-invariant haar-like feature in multiple targets [16]. In the study of target scattering characteristics using multi-aspect SAR, Zhao et al. analyzed the multiaspect scattering sensitivity characteristics, which can characterize the anisotropic scattering [17]. On this basis, Teng et al. used the idea of statistical fitting to give a quantitative analysis of anisotropy and isotropy, in which it was concluded that the $G^0$ distribution fitting SAR images is closer to the real data distribution than other distributions [18]. In addition, the use of multi-aspect scattering can distinguish man-made target from natural target in the scene [19]. In the above research process, the research point is mainly in target neighborhood (such as a $5 \times 5$ sliding window), and the anisotropy and isotropy are analyzed for a single point characteristics.

In this paper, on the one hand, the multi-aspect scattering curves of single point are analyzed, and the targets of different structures can be roughly analyzed based on the relative amplitude and the shape of the curve. On the other hand, slice extraction is first performed on the target, which contains target slices from all angles. Then, the $G^0$ distribution with statistical fitting is used to fit it, and the fitting estimation method is expectation maximization (EM) [20,21], which can more accurately estimate the corresponding SAR image parameters. The parameter-based multi-aspect SAR target image fitting has two parameters, and the parameter curve is drawn according to its angle. Finally, it is concluded that targets with different scattering characteristics (amplitude difference from surrounding points, scattering energy [22]) show different laws for the multi-aspect parameter distribution. Finally, based on the target slice, the influence of the sliding window sampling target on the parameters is discussed, and the analysis results of parameter model to represent different types of targets are given.

In the process of parameter solving, since it involves statistical distribution fitting, there is a moment estimation method and EM estimation method in the $G^0$ distribution parameter solution. The EM estimation method is not limited by the parameter size and the estimation result is more accurate [23]. The target is contained in the entire slice (for example, the size of $60 \times 60$ contains the entire target), and draws the curve distribution of the two parameters $\beta$ and $\sigma$ as a function of angle. The C-band circular SAR data contains many types of targets: warehouses, buildings, tanks and special structures. Based on the different backscattering of targets with different structures, the two parameters finally obtained show different distributions. In addition, we analyzed the experimental parameters of the window sampling target (for example, only partial structure of the warehouse appeared in the slice), compared the parameters changing with sliding window processing, and finally gave the parameter results of the target with different scattering types. The effectiveness of using the two parameter model to analyze target backscattering with different structures in multi-aspect SAR is shown.

In this paper, the Section 2 analyzes the statistical fitting method and parameter solution used in the experiment, and gives the operation steps of the experiment. The third part introduces the data used in the experiment and the results of the point target scattering analysis. In the fourth part, the two parameter solution of the target slice is carried out, and the corresponding experimental results are given. The fifth part is the conclusion.

## 2. Methodology and Experimental Procedures

### 2.1. Multi-Aspect SAR Observing Model

Multi-aspect SAR observation obtains echo data from different angles with changing of radar platform azimuth. The circular SAR is the complete data set of multi-aspect SAR, and the flight trajectory is carried out in a circle [24]. Figure 1 shows the simulation diagram of the flight platform, in which the red triangle represents the radar, and the blue circle simulates electromagnetic waves. The echo data is obtained by transmitting electromagnetic waves and collected at different angles.

There are targets with different structures in C-band multi-aspect SAR scene. The targets backscattering obtained through angles is different, and the scattering amplitude values between different targets will be quite different. We analyze objects with different structures such as warehouses, tanks, etc. in the C-band data used in the experiments. The multi-aspect data acquisition based on the observation model, on the one hand, fits the multi-aspect data of the target. On the other hand, the backscattering of different targets is compared, and the parameter model is used for further analysis on the basis of backscattering amplitude.

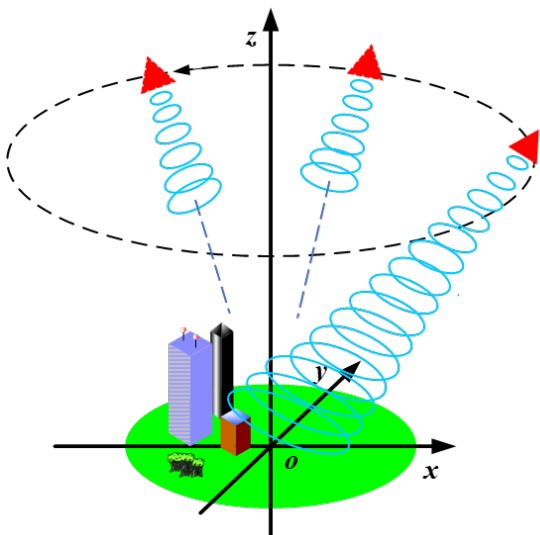

**Figure 1.** Circular SAR observation model.

### 2.2. Parametric Model

With the multi-aspect data acquisition, multi-aspect images are acquired through corresponding imaging algorithms. After slicing the targets in the scene, statistical fitting is performed on the target slices. The EM-based $G^0$ distribution is adopted to fit the targets (including one-dimensional backscattering curve analysis for point targets). The two parameters in the model correspond to the scattering characteristics of the target, respectively corresponding to the scattering amplitude and energy distribution of the target. There are typical parameter values in the parameter curve. Analyzing the scattering characteristics at corresponding angles can provide typical characteristics of the target along the angle, and further compare and analyze the differences in SAR target characteristics at different angles. The angle slices are fitted respectively to obtain the corresponding multi-aspect fitting parameters, and the parameters are used as the characterization method of target scattering analysis. After the entire target slice analysis, the target is sampled through the window, and the obtained parameters are used to characterize partial target.

#### 2.2.1. Distribution Function

Statistical fitting of SAR images, such as Rayleigh distribution, K distribution, $G^0$ distribution, etc. [25–27]. In [18], the distribution fitting is compared and analyzed, and it is concluded that the fitting effect of $G^0$ distribution is better than other distributions. At

the same time, in the process of $G^0$ distribution fitting, moment estimation is usually used to solve the parameters, but there is a problem of parameter limitation: $\beta > 2$ according to Equation (1). In our experiment, we use the EM iterative algorithm for parameter estimation. After EM solves the parameters, two parameters are obtained and named as $\beta$ and $\sigma$, which reflects the EM iterative equation. Estimating parameters is more accurate than moment estimation. In the experiment, these two parameters are used to analyze different targets, and it is found that the use of two parameters can represent targets with different scattering characteristics. The parameter beta is defined by the expression.

$$\beta = 1 + \frac{N\hat{m}_4}{N\hat{m}_4 - (N+2)\hat{m}_2^2} \tag{1}$$

$$\hat{m}_k = \frac{1}{T}\sum_{t=1}^{T} I_t^k \tag{2}$$

The second and fourth order moments of the amplitude data in [23] are used as equations to illustrate the moment estimation method. Where $N$ represents twice the equivalent number of sights. The equivalent number of views is determined before radar imaging as a parameter in the statistical fitting function. The process of statistical function representation produces twice the number of sights [23,28]. $\hat{m}_k$ represents the $k$-order sample moment, that is, the sum of the samples to the k-th power is averaged. $I_t$ is SAR real data, and the $T$ is number of data.

The commonly used $G^0$ distribution function is expressed as [29,30]:

$$p(I_t \mid \alpha, \gamma) = \frac{2n^n \Gamma(n-\alpha) I_t^{2n-1}}{\Gamma(n)\Gamma(-\alpha)\gamma^2 \left(\gamma + nI_t^2\right)^{n-\alpha}} \quad -\alpha, \gamma, n, I_t > 0 \tag{3}$$

where $n$ represents the equivalent number of sights, $\alpha$ is the shape parameter reflecting the measured area, $\gamma$ is the scale parameter and is related to the average energy of the measured area, $I_t$ is SAR real data.

A new form of $G^0$ distribution is given. From Bayesian theory, the $G^0$ distribution can be written as:

$$p(I_t \mid \lambda) = \int_0^\infty p(I_t \mid \omega_t) p(\omega_t \mid \lambda) \mathrm{d}\omega_t \tag{4}$$

The EM algorithm is based on maximizing the auxiliary function [31]:

$$\lambda = \underset{\omega}{\mathrm{argmax}} \int p(\omega \mid I, \lambda') \log p(I, \omega \mid \lambda) \mathrm{d}\omega \tag{5}$$

where $\lambda$ is parameter to be estimated, $\lambda'$ is current parameter estimates, $\omega$ is intermediate auxiliary variable. Assume that the variable $y_t$ is $N$ independent and identically distributed Gaussian random variables (mean is zero, and the variance is the square root of the sum of squares of $\omega_t$), then $p(I_t \mid \omega_t)$ conforms Generalized Rayleigh distribution:

$$p(I_t \mid \omega_t) = \frac{2I_t^{N-1}}{(2\omega_t)^{N/2}\Gamma(N/2)} \exp\left(\frac{-I_t^2}{2\omega_t}\right) \tag{6}$$

Assuming that the parameter $\omega_t$ conforms to the inverse Gamma distribution, there is:

$$p(\omega_t \mid \lambda) = \frac{\sigma^\beta}{\Gamma(\beta)} \omega_t^{-\beta-1} \exp\left(\frac{-\sigma}{\omega_t}\right) \tag{7}$$

In the Equation (7), $\lambda = \{\beta, \sigma\}$ is the parameter to be estimated. Putting Equations (6) and (7) into Equation (4), after integration, a new form of $G^0$ distribution can get:

$$p(I_t \mid \lambda) = \frac{2^{1+\beta} I_t^{N-1} \sigma^\beta}{\Gamma(N/2)\Gamma(\beta)} \left( I_t^2 + 2\sigma \right)^{-(N/2+\beta)} \times \Gamma(N/2 + \beta) \tag{8}$$

Compared with Equation (3), it is easy to get:

$$\begin{cases} N = 2n \\ \beta = -\alpha \\ \sigma = \gamma/N \end{cases} \tag{9}$$

2.2.2. Estimated Parameters

In the Equation (9), $n = 1$ according to the SAR prior, so the parameters to be estimated are $\beta$ and $\sigma$. Incorporate Equation (7) into Equation (5), take the right part of the equal sign to derive $\sigma$ and $\beta$ respectively and set them to 0, and finally get EM iterative formula (Refs. [31–33] for the specific derivation process):

Given the initial value of $\beta$ and $\sigma$, the result can be solved iteratively according to Equation (9). Where $\Psi()$ represents the digamma function.

$$\begin{cases} \beta_{(k+1)} = \frac{\left[ \ln \beta_{(k)} - \Psi\left(\beta_{(k)}\right) \right]}{\ln(AG)} \beta_{(k)} \\ \sigma_{(k)} = \frac{\beta_{(k)}}{A} \\ A = \frac{1}{T} \sum\limits_{t=1}^{T} \frac{N + 2\beta_{(k)}}{I_t^2 + 2\sigma_{(k)}} \\ G = \left( \prod\limits_{t=1}^{T} \left( \frac{I_t^2 + 2\sigma_{(k)}}{2} \right) \right)^{1/T} \exp\left( -\Psi\left( \frac{N + 2\beta_{(k)}}{2} \right) \right) \end{cases} \tag{10}$$

Each SAR image corresponds to a pair of two parameter solution results. For example, the first sub-aperture target slice, after the iteration of the above formula, the corresponding double parameters $\left\{ \beta^{(1)}, \sigma^{(1)} \right\}$ are obtained, and after multi-aspect $\{a_1, a_2, a_3 \ldots a_n\}$, the parameter results are as follows:

$$\begin{aligned} P_0 &: \{\beta_{a_1}, \beta_{a_2}, \ldots, \beta_{a_n}\} \\ P_1 &: \{\sigma_{a_1}, \sigma_{a_2}, \ldots, \sigma_{a_n}\} \end{aligned} \tag{11}$$

The set of two parameters is $\{P_0, P_1\}$, and the two parameter curve of the target can be obtained through the above formula. When multiple targets are selected for analysis, the two parameter results of the corresponding targets are obtained: $\{P_{k0}, P_{k1}\}$, where $k$ represents the $k_{th}$ target. Therefore, the corresponding parameter curve can be obtained from the multi-aspect slice images of each target. In the experiment, four targets: warehouse, tank, building, and special structure were collected to analyze the parametric model, so $k = 4$. Among them, the structure of the warehouse is square and has a symmetrical structure, and the structure of the tank is circular. Scattering characteristics show strong backscattering amplitude at a certain angle [34,35]; buildings and special structures are analyzed as two complex structures.

Figure 2 shows the solution flow diagram of the above formula, which is roughly divided into four parts. First, the warehouse area is selected, multi-aspect slicing is performed on it, and then the statistical model is used for fitting and solving, and finally the two parameters of the area are obtained.

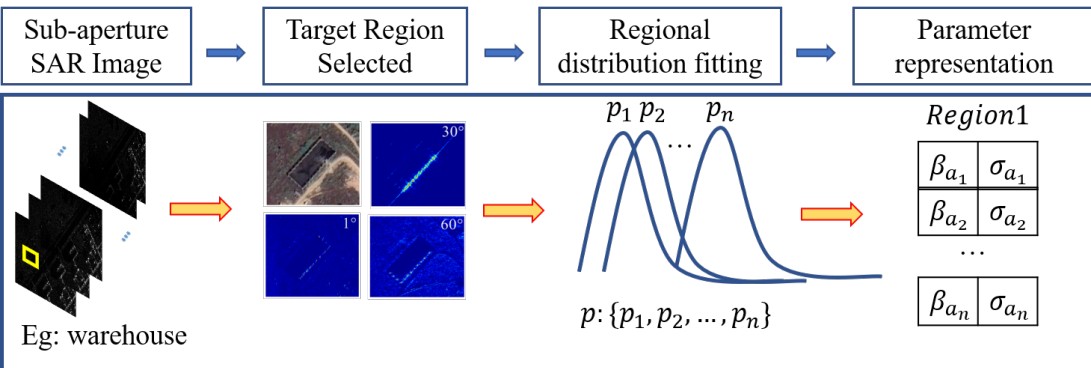

**Figure 2.** Schematic diagram of parameter characterization.

The two parameters' curve is obtained on the premise that the entire target is fixed in the slice. In actual situations, the image sampled by the window contains the part of the target but not all the information. We collect the slices of the target with window, and solve the parameters of the obtained multi-slices respectively. Assuming that the $k_{th}$ target is analyzed, first extract the target slice (the size of the slice is given in the experimental results section), set the target slice as $s_1, s_2, s_3, \ldots, s_i$, and perform a two parameter solution for each slice, and finally get the two parameters of the target as follows:

$$P_{k0}: \left\{ \begin{array}{c} \beta_{a_1}^{(s_1)}, \beta_{a_2}^{(s_1)}, \ldots, \beta_{a_n}^{(s_1)} \\ \beta_{a_1}^{(s_2)}, \beta_{a_2}^{(s_2)}, \ldots, \beta_{a_n}^{(s_2)} \\ \vdots \\ \beta_{a_1}^{(s_i)}, \beta_{a_2}^{(s_i)}, \ldots, \beta_{a_n}^{(s_i)} \end{array} \right\} \tag{12}$$

$$P_{k1}: \left\{ \begin{array}{c} \sigma_{a_1}^{(s_1)}, \sigma_{a_2}^{(s_1)}, \ldots, \sigma_{a_n}^{(s_1)} \\ \sigma_{a_1}^{(s_2)}, \sigma_{a_2}^{(s_2)}, \ldots, \sigma_{a_n}^{(s_2)} \\ \vdots \\ \sigma_{a_1}^{(s_i)}, \sigma_{a_2}^{(s_i)}, \ldots, \sigma_{a_n}^{(s_i)} \end{array} \right\} \tag{13}$$

It can be seen from the above formula that each target can obtain a two parameter formula $\{P_{k0}, P_{k1}\}$, and the rows and columns in the formula can be analyzed separately. In the experiment, the four kinds of targets were sliced, and the parameter changes were observed through sampling. The multi-aspect targets were characterized by two parameters combining with backscattering amplitude. In Section 4, the parameter values of the target with the angle and the two parameter comparison between different targets, such as the relative magnitude of the parameter value and the shape of the parameter curve, are carried out by using the two parameter model.

### 2.3. Experiment Procedure

Figure 3 shows the general framework of the experiment. The first part is the preprocessing of the data: a low-rank matrix decomposition based on the robust principal component analysis solution is used [36–38], and the preprocessing is not analyzed in detail in this experiment. The second part compares the backscattering of different point targets, including fitting the target SAR image. The third part is divided into two parts for parameter extraction: a slice containing the overall target, and a slice containing partial target. For the former, a parameter curve about the target is obtained, and for the latter, a parameter matrix of the target slice location is obtained. The fourth part aims to use the parameters obtained in the previous part to carry out multi-aspect scattering comparison of the target and the comparison between targets, and finally give the range of parameter values that characterize the target scattering.

Experimental processing is based on the structure of the target. In [19], previous work used sliding windows to collect data without considering the structure of the target. In the experimental processing, slices containing the overall target structure were extracted, and partial structures (such as a quarter) were selected for parameter analysis. The experimentally extracted structure can map quantitative parameters to the target structure.

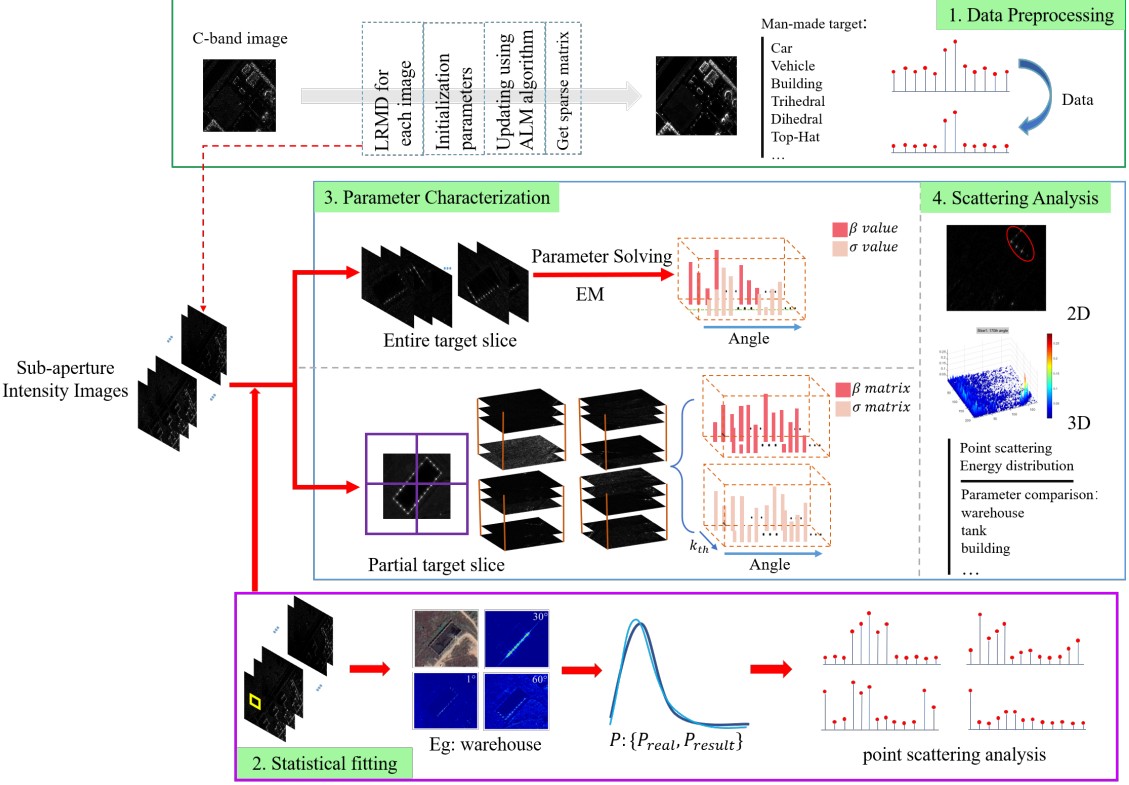

**Figure 3.** The detailed implementations of using algorithm.

## 3. Dataset and Scattering Analysis

### 3.1. Dataset

The data is a C-band circular SAR dataset. It was obtained by the Aerospace Information Research Institute, Chinese Academy of Sciences (AIRCAS) in Zhuhai, Guangdong Province, China. Table 1 shows the experiment parameters. Figure 4 shows optical image of observing scene. Figure 5 shows the 360° coherent complex image using BP algorithm [5,39,40].

**Table 1.** Parameters of the C-band circular SAR experiment.

| Parameters | Value |
| --- | --- |
| Carrier frequency | 5.4 GHz |
| Bandwidth | 560 MHz |
| PRF | 2358 Hz |
| Flight height | 3 km |
| Flight radius | 5 km |
| Range resolution | 0.27 m |
| Azimuth resolution | 0.3 m |

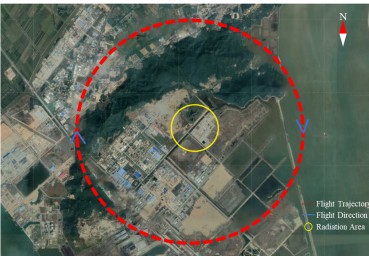

**Figure 4.** Optical image of the observing scene and airplane flight trajectory.

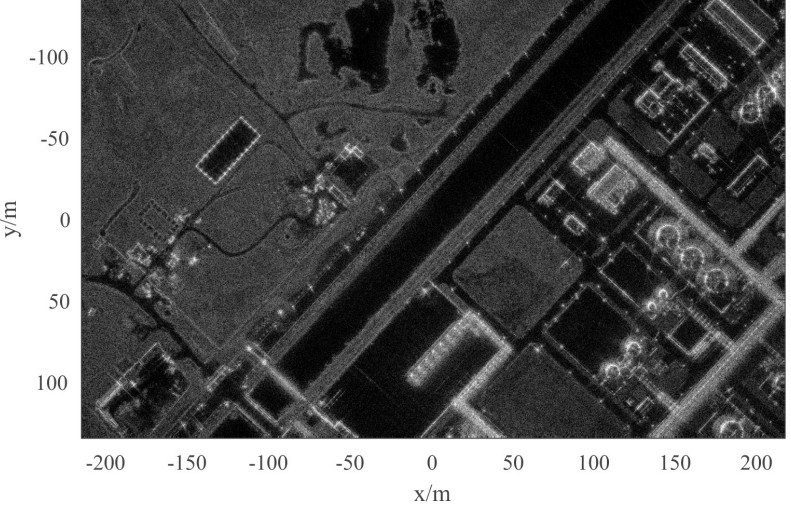

**Figure 5.** Coherent complex image of C−band data.

### 3.2. Scattering and Statistical Analysis

In the experiment, warehouses, tanks, buildings, and special structures in Figure 4 were selected for analysis. The optical and full-aperture slices of the corresponding targets are shown in Figure 6. A point was selected from Figure 6 for backscattering analysis (marked by arrows in the figure). Combining SAR images and optical images, it can be seen that (a) (b) (d) three types of targets have less surrounding interference (metal object interference), and the selected tank is the first one in the upper left corner of (b), and selected building in (c) top right, it can be seen that there are small tanks, metal racks and other objects around it.

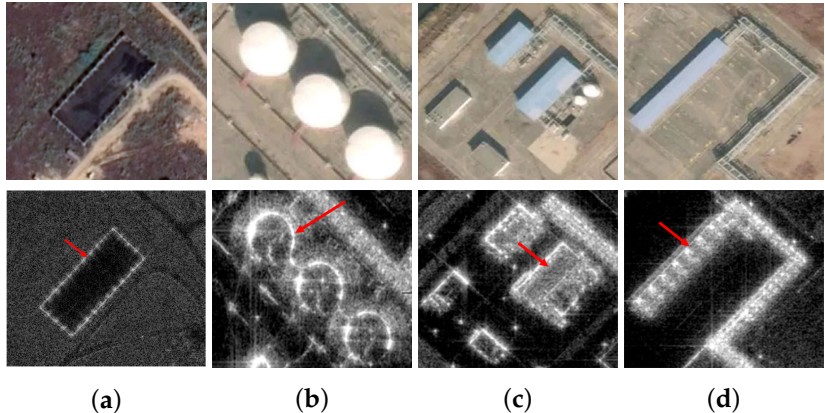

|     |     |     |     |
| --- | --- | --- | --- |
| (**a**) | (**b**) | (**c**) | (**d**) |

**Figure 6.** Optical images of four man-made targets and corresponding SAR images: (from left to right) (**a**) The optical image and SAR image of warehouses; (**b**) The optical image and SAR image of tanks; (**c**) The optical image and SAR image of buildings; (**d**) The optical image and SAR image of special structures.

One-dimensional backscattering curve analysis is carried out by selecting one point for the four targets respectively, as shown in Figure 7, the one-dimensional backscattering curve of four points is given. The red box circles strong scattering area of the target point. For targets with regular structures such as warehouses and tanks (it can be seen from the optical image that the special structures are also regular). Strong scattering occurs in the backscattering amplitude curve is a certain angle range. For example, a strong response in the 200–230 angle range for the warehouse point. However, due to the existence of small targets such as iron frames around the building, the backscattering curve of the research point may be affected by multiple reflections [10,41], so it can be seen from Figure 7c that four strong response areas are drawn. At the same time, from the red box of the curve, we can see the difference in the backscattering amplitude between different targets: the backscattering amplitude of the tank is close to 1300 at most, followed by warehouses, special structures, and buildings.

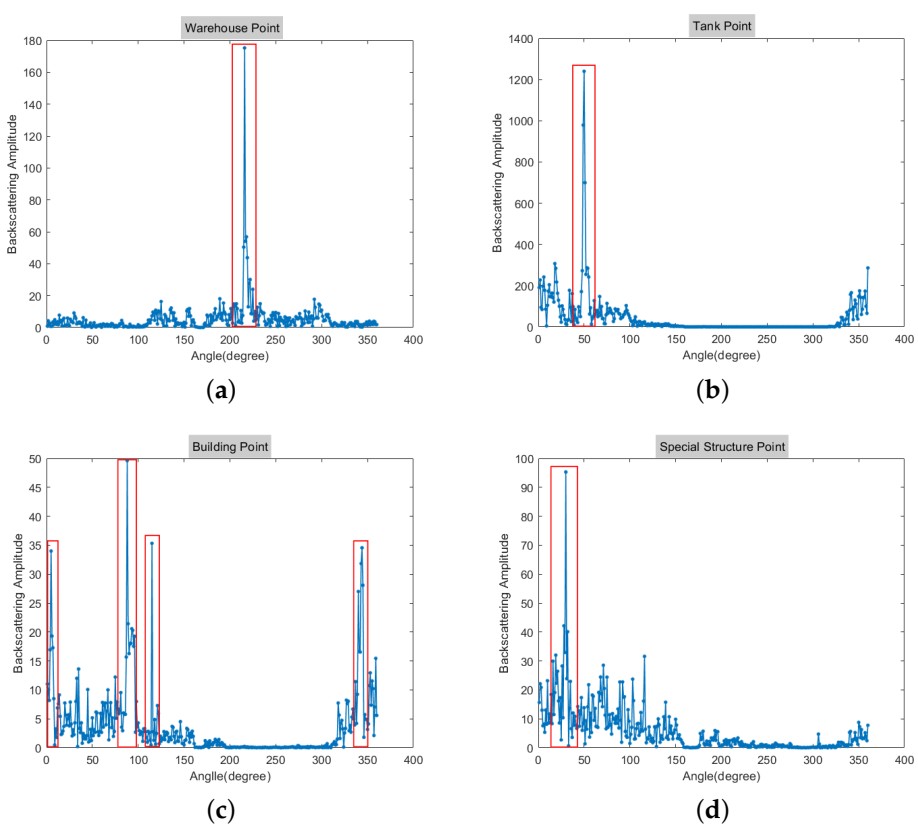

**Figure 7.** One-dimensional backscattering curve distribution: (**a**) represents backscattering amplitude curve of warehouse point; (**b**) represents backscattering amplitude curve of tank point; (**c**) represents backscattering amplitude curve of building point; (**d**) represents backscattering amplitude curve of special structure point.

In [19], the curve and parameters of the statistical distribution have been studied, so two targets are used for statistical display here. As shown in Figure 8 the statistical curves of warehouses and tanks are given. Compared with the original data, this statistical method can closely represent the SAR target image. The backscattering of the target is preliminarily analyzed with the above, and the next part is the parameter representation of the multi-aspect target slice image, and parameterized representation of the multi-aspect target is obtained. One-dimensional point scattering roughly shows the change of the point's response to radar. At the same time, the point may be affected by the surrounding multiple scattering. The point scattering curve analysis cannot represent the type of the target, and even if neighborhood information is added, it cannot reflect the overall multi-aspect changing of the target's scattering or target's image. On the basis of statistical fitting,

it is necessary to use parameters to analyze the multi-aspect target slices. On the one hand, the multi-aspect target scattering is extracted by parameterization, and on the other hand, the parameters of with different structures are compared.

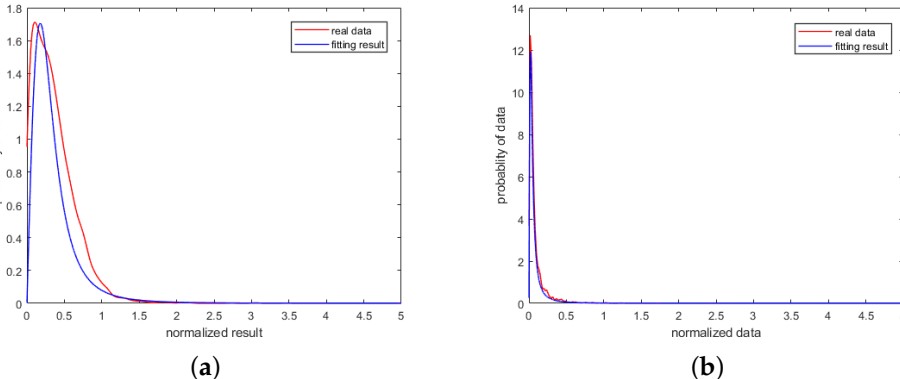

**Figure 8.** Statistical Fitting Results: (**a**) Data curve and fitting curve of warehouse at 1st angle; (**b**) Data curve and fitting curve of tank at the 100th angle.

## 4. Parametric Characterization

### 4.1. Target Parameter Result I

The multi-aspect slices of the four targets in Figure 5 are selected, and two parameters are calculated by using Equation (10), and the obtained two parameters are used to characterize targets with different structures. Figure 9 shows the $\beta$ values of the four multi-aspect target slices. From the parameter distribution, it can be seen that the warehouse has obvious parameter distribution. On the one hand, the maximum value of the parameter is larger than that of the other three targets (similar to symmetrical features). Secondly, for the tank a larger $\beta$ appears around the 199th angle than other angles. For the other two targets, there will also be a slightly larger $\beta$ value at some angles, but generally its value is between 0.5 and 1. From the result of the four kinds of targets, it can be seen that the values of targets with different structures change differently. Using the $\beta$ value alone can qualitatively reflect the radar's response to different target. For example, for a warehouse at the 170th angle, it can be guessed that the response of the warehouse at this angle is very different or small from the response value of the surrounding environment, which is a response turning point. For the latter two objectives such as buildings, it can be seen that the solution value does not fluctuate greatly.

In order to quantitatively describe the parameters corresponding to the changes in the scattering intensity of the target image, two targets, warehouse and tank, were selected for analysis. Taking the 170th angle image of the warehouse and the 199th angle image of the tank as the main (the two angles with the largest $\beta$ value), other three angles are selected for comparison. Taking the multi-aspect warehouse slice in Figure 10 as an example (the 73rd, 170th, 206th, and 287th angle images were selected for comparison). From the comparison of strong and weak scattering between warehouse points and surrounding target points, it can be seen that the backscattering of the target at the 170th angle is closer to the surrounding environment of the target, and its $\beta$ value is at most 5.518 in all angles. Figure 11 shows the comparison curve of strong and weak points backscattering(selecting the amplitude values of target points and nearby non-target points). For the difference between tank target at the 199th angle and other slices, its $\beta$ value is at most 2.255 at this angle. From the SAR image comparison, the relationship between parameter and the backscattering of the target point can be qualitatively analyzed. The larger the $\beta$ value is at a certain angle, the closer the backscattering of the target point is to the backscattering of the surrounding environment. The $\beta$ value is small at a certain angle, and the backscatter of the target point is quite different from the backscatter of the surrounding environment.

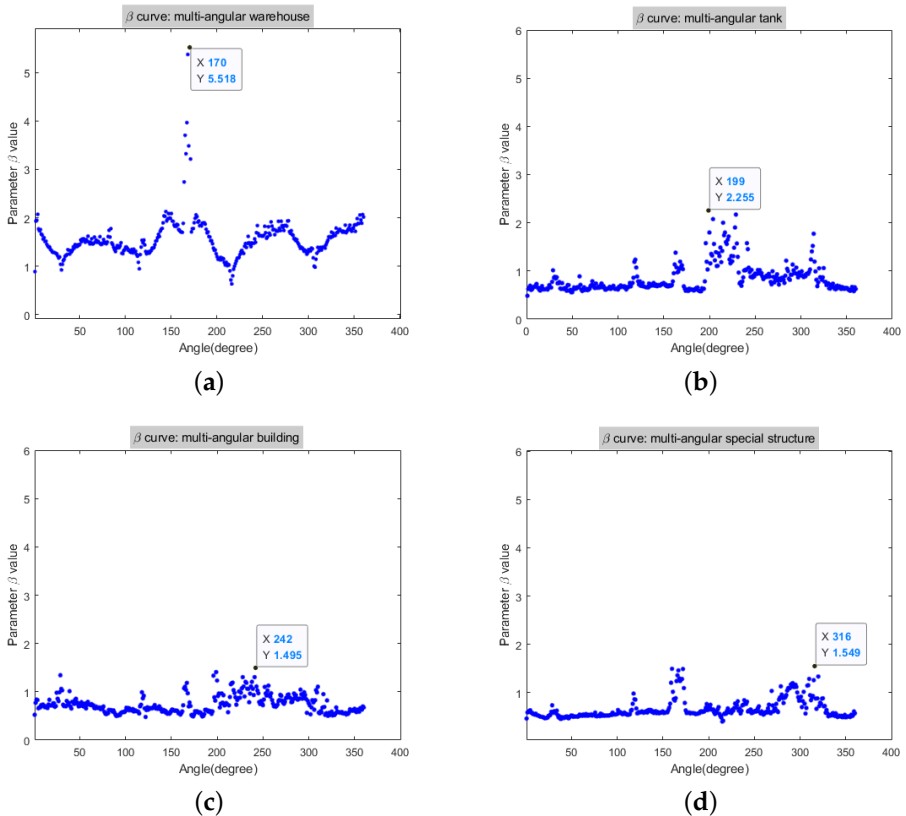

**Figure 9.** The $\beta$ parameter values of the four multi-aspect target slices: (**a**) represents $\beta$ value of multi-aspect warehouse slice; (**b**) represents $\beta$ value of multi-aspect tank slice; (**c**) represents $\beta$ value of multi-aspect building slice; (**d**) represents $\beta$ value of multi-aspect special structure slice.

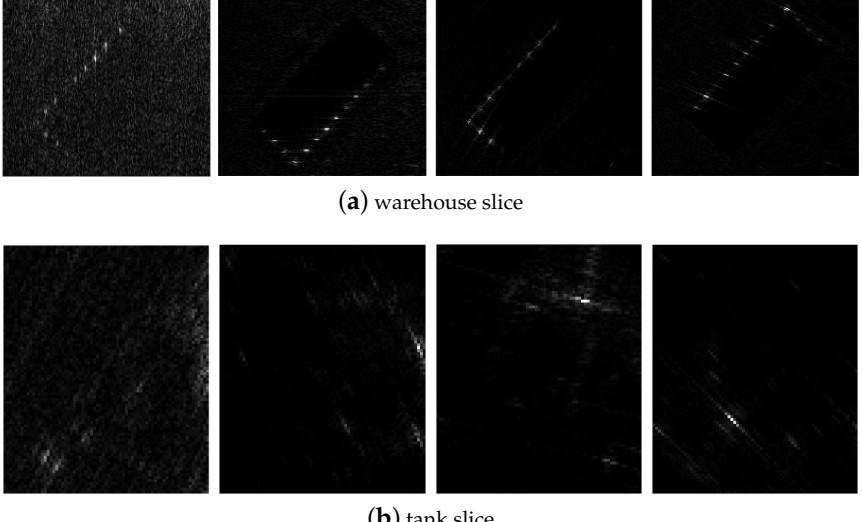

**Figure 10.** Four angle target slices for warehouse and tank: (**a**) represents 170th, 73rd, 206th, 287th angle warehouse slice; (**b**) represents 199th, 143rd, 92nd, 303rd angle tank slice.

Figure 11 aims at the quantitative analysis of the multi-aspect SAR images corresponding to the two targets. The four images of two targets in Figure 10 are used for analysis, respectively. The backscattering amplitude values of the target point and the surrounding points are selected for comparison. Figure 11a shows the amplitude values of the target point and the surrounding points in the warehouse. The red dotted circle shows amplitude of the two points at the 170th angle. It can be seen that the backscattering amplitude values

of the target point and surrounding points are close at this angle, while the parameter $\beta$ value appears to be the largest. For the remaining three angles, the backscattering amplitude value of the target point is large, and the corresponding parameter $\beta$ value is small. Similarly, the tank point at the 199th angle is close to the backscattering amplitude value of the surrounding points, and the value is the largest. The backscattering amplitude at the remaining three angles is quite different (the value is small).

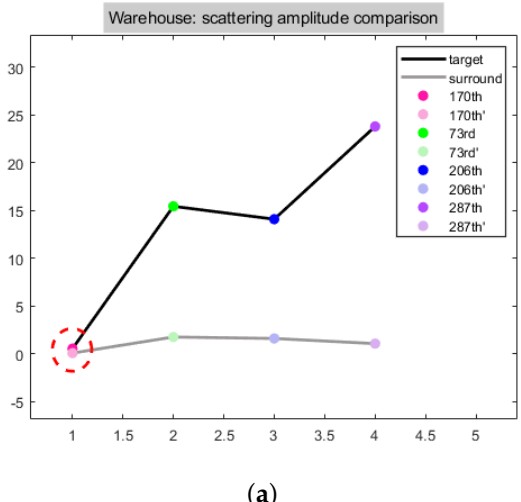

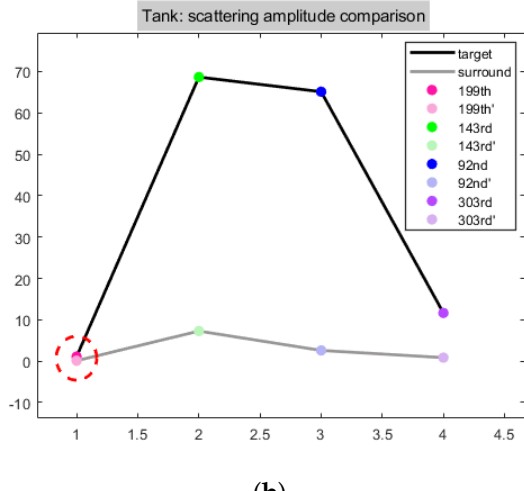

(**a**)                                                                                                                  (**b**)

**Figure 11.** Comparison of scattering amplitude between target point and surrounding points: (**a**) Warehouse:170th, 73rd, 206th, 287th angle (' represents surrounding point); (**b**) Tank:199th, 143rd, 92nd, 303rd angle. (The red circle represents the lowest value for difference between target point and surrounding point)

On the basis of solving the parameter $\beta$, the scattering difference between the target point and surrounding points can be obtained. In the parametric model, there is also a $\sigma$ parameter. From the following experimental results, it can be concluded that there are differences in the $\sigma$ curves for targets. Figure 12 shows the $\sigma$ solution results of the four targets. The $\sigma$ curve can be used to distinguish the targets of four different scattering types. The red part circles the $\sigma$ worth highlighting part. It can be seen that the four kinds of targets can be distinguished.

For the warehouse, the characteristics of the $\sigma$ curve are not obvious compared to the characteristics of the $\beta$ curve, and the prominent part is not much different from the surrounding values. For the two targets: tank and building, three protruding parts are circled, probably around the 10th, 120th, and 355th angles, but the solution values are different. It can be seen from Figure 12b that the highest value in the 10th angle range is close to 10, and the values in the remaining two ranges are around 3–5. For buildings, the values in the first highlighted range are smaller than the second red circle, while the third range has a minimum value around 1–2. For the fourth target, two solution ranges are circled, which have obvious characteristics compared with other targets.

### 4.2. Target Parameter Result II

#### 4.2.1. Parameter $\beta$

In order to have the randomness of the target selection part, a sliding window is used to select the target part slicing in the warehouse. Because the pixel size of the above slices is 180 × 200 (the four neighbors of the target are studied [42,43]), the sliding window step is 90 × 100, and four kinds of slices centered on the target are selected for parameter calculation (all include the warehouse part). Taking the $\beta$ parameter as an example, the multi-aspect parameter results of the four slices are shown in Table 2 (for the display, the 14-point angle results are taken, and the 170th angle is included as a comparative analysis)

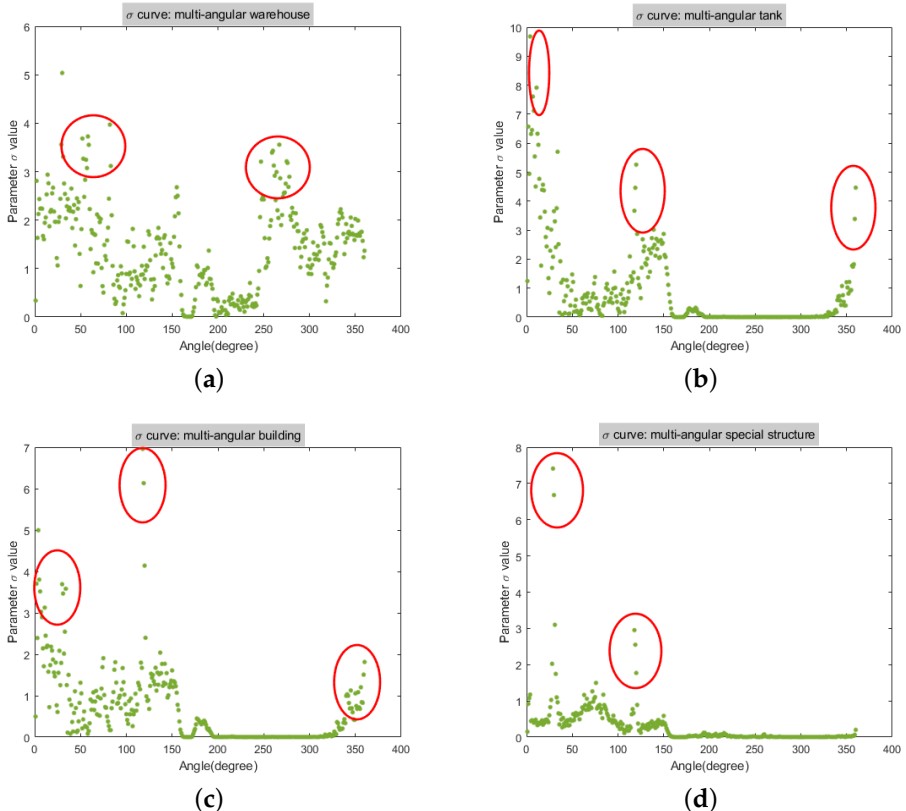

**Figure 12.** The $\sigma$ multi-aspect distribution of the four targets and the key areas are marked: (**a**) the $\sigma$ distribution of the warehouse and the two salient areas; (**b**) the $\sigma$ distribution of tank and the three salient areas; (**c**) the $\sigma$ distribution of the building and three salient areas; (**d**) The $\sigma$ distribution of the special structure and two salient areas.

**Table 2.** $\beta$ parameter: four kinds of slices centered on the warehouse for 14 angles (Bold indicates the maximum value).

| Selected Angle | Slice1 | Slice2 | Slice3 | Slice4 |
|---|---|---|---|---|
| 50th | 4.0027 | 2.0413 | 1.7205 | 2.2264 |
| 100th | 3.3064 | 1.8641 | 1.5675 | 1.6111 |
| 150th | 3.7451 | 2.6033 | 2.2140 | 2.6689 |
| 167th | 7.4318 | **9.7013** | 3.8194 | 6.4576 |
| 168th | 4.7545 | 3.0603 | 4.6331 | 9.0546 |
| 169th | 2.8312 | 2.0429 | 5.3442 | **13.1805** |
| 170th | 3.8806 | 2.3414 | **5.6539** | 12.3468 |
| 171st | 3.2174 | 2.1953 | 4.6044 | 8.5438 |
| 172nd | 3.0776 | 2.0824 | 2.0654 | 2.3499 |
| 173rd | 3.2524 | 1.9992 | 1.9934 | 2.0072 |
| 200th | 2.7945 | 2.3107 | 1.7552 | 2.1728 |
| 250th | 3.8236 | 2.1811 | 2.3832 | 2.6348 |
| 300th | 2.9585 | 1.6003 | 1.8814 | 2.5514 |
| 350th | **7.5452** | 2.5207 | 2.3703 | 2.3482 |

The solution for 14 angles are given in the table, and three angles are selected around the 170th angle. Slice1 represents the upper left of the window; Slice2 represents the upper right of the window; Slice3 represents the lower left of the window; Slice4 represents the lower right of the window. The maximum value for Slice1 is the 350th angle in the table, and the remaining three slices are at the 167th, 170th, and 169th. In fact, the parameter value 7.4318 for Slice1 at the 167th angle is close to the 350th corresponding value of 7.5452.

Therefore, after the sliding window selection, the $\beta$ value of the slice containing only part of the target also shows obvious characteristics around the 170th angle. The SAR images were also quantitatively compared and analyzed. The 167th, 167th, 170th, and 169th slices were selected as the main comparison for the above four slices, and the remaining three angle slices were selected for auxiliary comparison (a total of 16 SAR slices).

It can be seen from Figure 13 that the four slices respectively contain partial warehouse, and there is no interference from other targets in the surrounding. The first column is the larger parameter value corresponding target slice, and the approximate position of the target is drawn in green. The remaining three columns are the target slice corresponding to the smaller parameter solution, and the approximate position of the target is drawn in red. It can be seen in SAR image from the qualitative analysis that the target point with larger parameters is similar to the backscattering of the surrounding points, and it is even difficult to distinguish. The slices with smaller parameters have different backscattering between the target point and the surrounding points, and the structure of partial target can be seen. Similar to the analysis in the previous section, the target point in each slice and the surrounding points are selected for quantitative comparison. Table 3 shows the comparison of the backscattering amplitude.

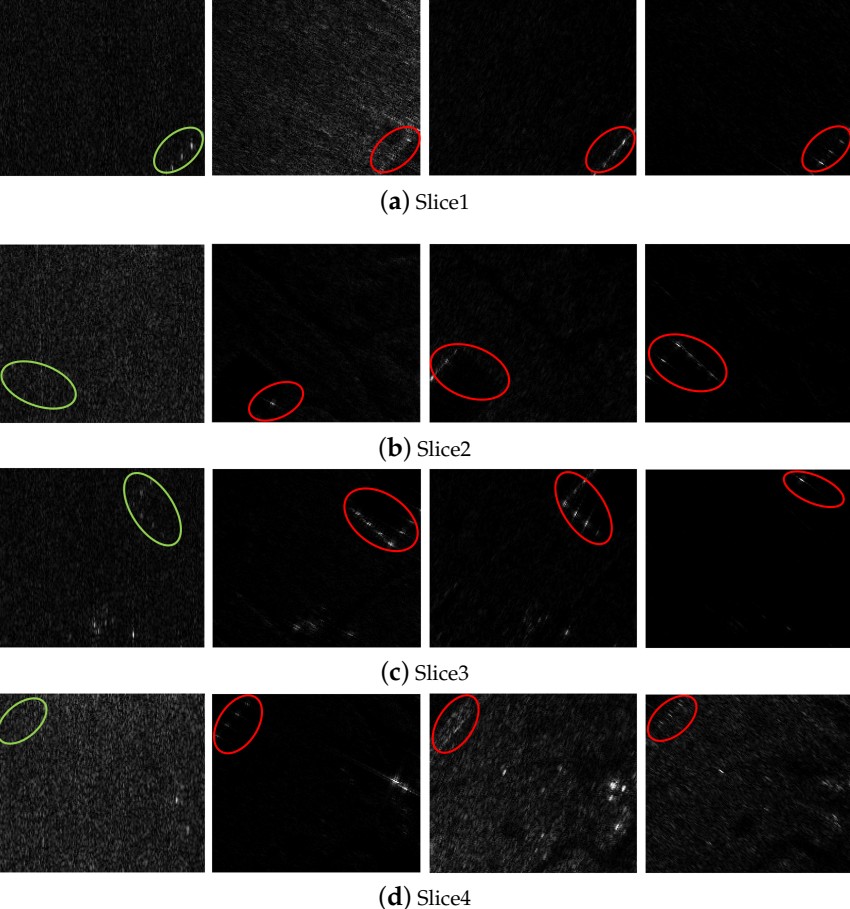

**Figure 13.** Backscattering comparison of multi-aspect slices containing different warehouse parts: (**a**) Slice1: Multi-aspect slice containing the upper left part of the warehouse (167th, 100th, 200th, 300th angle slices from left to right); (**b**) Slice2: Multi-aspect slice containing the upper right part of the warehouse (167th, 100th, 200th, 300th angle slices from left to right); (**c**) Slice3: Multi-aspect slice containing the lower left part of the warehouse (170th, 100th, 200th, 300th angle slices from left to right); (**d**) Slice4: Multi-aspect slice containing the lower right part of the warehouse (169th, 100th, 200th, 300th angle slices from left to right).

For backscattering amplitude in Table 3, each angle corresponds to two points data, namely the backscattering amplitude of the target point and the surrounding points (select the target point in the circled part of Figure 13). The bold part in Table 3 corresponds to the angle with the larger parameter, and it can be seen that the upper and lower are close. Taking Slice1 as an example, the two data at the 167th angle are 0.8057 and 0.1181 and the difference is 0.6876. The differences at the remaining three angles are all greater than this value. Corresponding to the law of $\beta$ value research in the study, the larger the parameter is, the closer the target point is to the surrounding points, otherwise there is a large difference.

**Table 3.** Backscattering amplitude comparison: target point and surrounding point selected in Figure 13 * (100th, 200th, 300th angle as comparative analysis).

| Slice1 | | Slice2 | | Slice3 | | Slice4 | |
|---|---|---|---|---|---|---|---|
| 167th | **0.8057** **0.1181** | 167th | **0.08114** **0.01625** | 170th | **0.2568** **0.02183** | 169th | **0.02839** **0.01483** |
| 100th | 1.644 0.7367 | 100th | 32 0.4394 | 100th | 18.34 0.337 | 100th | 13.27 0.3396 |
| 200th | 21.6 1.393 | 200th | 9.943 0.3437 | 200th | 23.76 0.6441 | 200th | 4.326 0.2995 |
| 300th | 43.39 1.163 | 300th | 29.98 0.4281 | 300th | 37.44 0.5717 | 300th | 11.82 0.7429 |

* The amplitude of the target point is up.

### 4.2.2. Parameter $\sigma$

Another parameter in the parametric model: $\sigma$, was analyzed in the previous section, and the target was simply analyzed based on the characteristics of the curve. In this section, similar to the $\beta$ parameter above, the target is quantitatively analyzed through the solution value. Through experimental analysis, the $\beta$ value represents the relative change of the target's scattering or target's image, and is independent of the amplitude itself. The $\sigma$ value is solved through the following experiment and found that the $\sigma$ value is related to the energy of the target point (in the experiment, the square of the backscattering amplitude represents the energy).

Table 4 shows the corresponding angle's $\sigma$ value of the four slices. Similar to the $\beta$ value analysis, the man angle is 170th (three angles from the left and right added). The six surrounding angles including the 170th angle are used as experimental angles, with an interval of 50 degrees starting from the 50th angle as a comparison. Both Slices1 and 2 have the smallest values at the 170th angle: 0.0107 and 0.0086. Both Slice3 and 4 have the smallest values at the 167th angle: 0.0056 and 0.0092. The values around 170th for the four slices are small compared to the other angles. The calculated value of the parameter is related to the energy distribution of the region, and the calculated value of the parameter around the 170th angle corresponds to the minimum value of the maximum energy with different angles. The $\beta$ characterizes the difference in the backscattering amplitude between the target points and the surrounding points. The experiment uses the square of the amplitude to represent the energy, and combines the parameter $\sigma$ to analyze the target slice. As shown in Figure 14, the target energy distribution and energy scale of Slice1 are given. From Figure 14a, it can be seen that the maximum value of the target point is 0.28, and the highest values of remaining three slices are about 60, 1400, and 11,000. It can be preliminarily concluded that the smaller $\sigma$ value, the weaker backscattering energy of target, and the larger $\sigma$ value, the stronger backscattering energy of target.

**Table 4.** $\sigma$ parameter: four kinds of slices centered on the warehouse for 14 angles (Bold indicates the minimum value; add angles for analysis in 170th angle).

| Selected Angle | Slice1 | Slice2 | Slice3 | Slice4 |
|---|---|---|---|---|
| 50th | 1.7803 | 1.8528 | 0.5084 | 1.9273 |
| 100th | 1.1458 | 1.9270 | 0.8918 | 2.4094 |
| 150th | 1.4322 | 1.2189 | 1.8844 | 2.6954 |
| 167th | 0.0114 | 0.0144 | **0.0056** | **0.0092** |
| 168th | 0.0118 | 0.0086 | 0.0069 | 0.0124 |
| 169th | 0.0114 | 0.0101 | 0.0073 | 0.0183 |
| 170th | **0.0107** | **0.0086** | 0.0077 | 0.0173 |
| 171st | 0.0122 | 0.0119 | 0.0064 | 0.0135 |
| 172nd | 0.1081 | 0.0917 | 0.0120 | 0.0203 |
| 173rd | 0.4637 | 0.3360 | 0.0554 | 0.0809 |
| 200th | 1.2545 | 0.2853 | 1.4792 | 0.7440 |
| 250th | 10.7151 | 2.6559 | 5.3388 | 3.8337 |
| 300th | 4.3230 | 1.6239 | 1.6452 | 1.6228 |
| 350th | 12.3619 | 3.6842 | 2.9505 | 3.0283 |

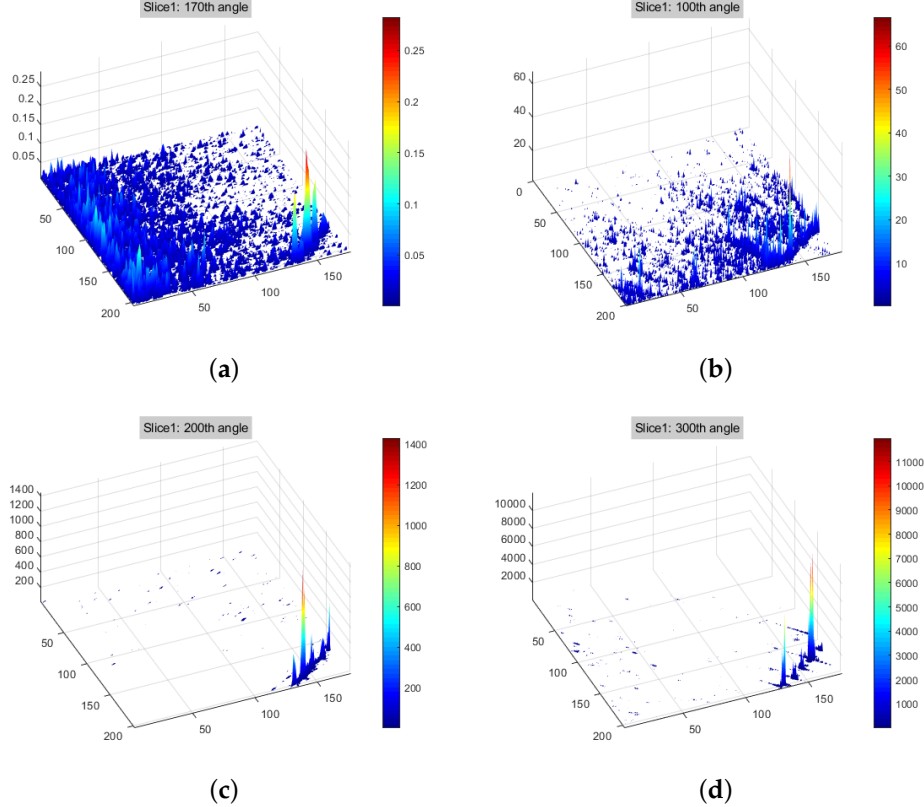

**Figure 14.** Target energy distribution for Slice1 (partial warehouse exists in slice): (**a**) 170th angle energy distribution: $\sigma = 0.0107$, maximum energy about 0.28; (**b**) 100th angle energy distribution: $\sigma = 1.1458$, maximum energy about 60; (**c**) 200th angle energy distribution: $\sigma = 1.2545$, maximum energy about 1400; (**d**) 300th angle energy distribution: $\sigma = 4.3230$, maximum energy about 11,000.

Table 5 gives the solution results of the $\sigma$ and the maximum energy (square of the backscattering amplitude) for the four slices (upper left, upper right, lower left, lower right) of the warehouse. As can be seen from the table, the $\sigma$ value in the 170th angle slice obtained is much smaller than the other three angle slices, and the corresponding maximum energy is also much smaller than the other three angle slices. From a lateral perspective, as the parameter increases, the corresponding energy value also increases. The 300th angle of Slice1 reaches $1.076 \times 10^4$. Quantitatively and qualitatively, the use of parameters to

characterize the energy of some target slices is in line with the actual situation. In order to increase the contrast of various targets, the upper left slice of remaining three targets was selected to conduct a comparison experiment from $\sigma$ parameters to energy distribution.

**Table 5.** Four slices of warehouse comparison from angle and value: minimum $\sigma$ value appeared around 170th angle (100th, 200th, 300th angle as comparative analysis ).

| Parameters | Slice1: 170th angle | Slice1: 100th angle | Slice1: 200th angle | Slice1: 300th angle |
|---|---|---|---|---|
| $\sigma$ | **0.0107** | 1.1458 | 1.2545 | 4.3230 |
| Maximum Energy | **0.2824** | 66.71 | 1430 | $1.076 \times 10^4$ |
| | **Slice2: 170th angle** | **Slice2: 100th angle** | **Slice2: 200th angle** | **Slice2: 300th angle** |
| $\sigma$ | **0.0086** | 1.9270 | 0.2853 | 1.6239 |
| Maximum Energy | **0.4417** | $2.524 \times 10^4$ | 459.4 | 5451 |
| | **Slice3: 170th angle** | **Slice3: 100th angle** | **Slice3: 200th angle** | **Slice3: 300th angle** |
| $\sigma$ | **0.0077** | 0.8918 | 1.4792 | 1.6452 |
| Maximum Energy | **0.8304** | 5425 | 5976 | $8.358 \times 10^4$ |
| | **Slice4: 170th angle** | **Slice4: 100th angle** | **Slice4: 200th angle** | **Slice4: 300th angle** |
| $\sigma$ | **0.0173** | 2.4094 | 0.7440 | 1.6228 |
| Maximum Energy | **0.1128** | $2.408 \times 10^5$ | 430.4 | 597 |

Figure 15 shows the slice image of tank, building, and special structure and the energy distribution corresponding to the $\sigma$ value. According to the calculation results of the parameters, the corresponding angle SAR image is selected to describe its energy distribution. Figure 15 shows the corresponding relationship between typical parameter values and the energy distribution diagram. The target data selected from the SAR image is a quarter of the target, and the partial structure containing the target is subjected to parameter extraction and scattering feature analysis. From the SAR image, it can be seen that some information of corresponding target is included, for example, only partial tank exists with a ring shape. The second column selects a larger $\sigma$ value and does not select the largest value for analysis: on the one hand, combined with the $\sigma$ curve of the overall target, it is obtained that the maximum value is not represented in the 100th–150th angle range, and on the other hand, after analysis with maximum value it is found that at this angle, because this part of the target's scattering or target's image shows weak scattering, while the nearby small targets show strong scattering, so the final value is larger. Taking the tank as an example, the maximum value appears around the 130th angle. In order to avoid the above situation, the value of the 34th angle is selected. It can be seen that the energy is concentrated on the target, and the maximum value is about $6.5 \times 10^5$. For special structure, since the target occupies a large number of pixels and there is no interference from other targets in the slice, the solved value is obvious so the maximum value is selected for analysis. The third column is the minimum value of the selected parameters for comparative analysis. Take the tank as an example: the minimum value appears at the 203rd angle. It can be seen that the energy also has a clear distribution, but the value is small and about 5. At the same time, the energy does not belong to the target, and can be ignored compared to the energy of target. For special structure, since the slice contains many parts, there are also targets in the energy distribution with the smallest value, but its energy value about 1.7 is much smaller than the value corresponding to the 30th angle.

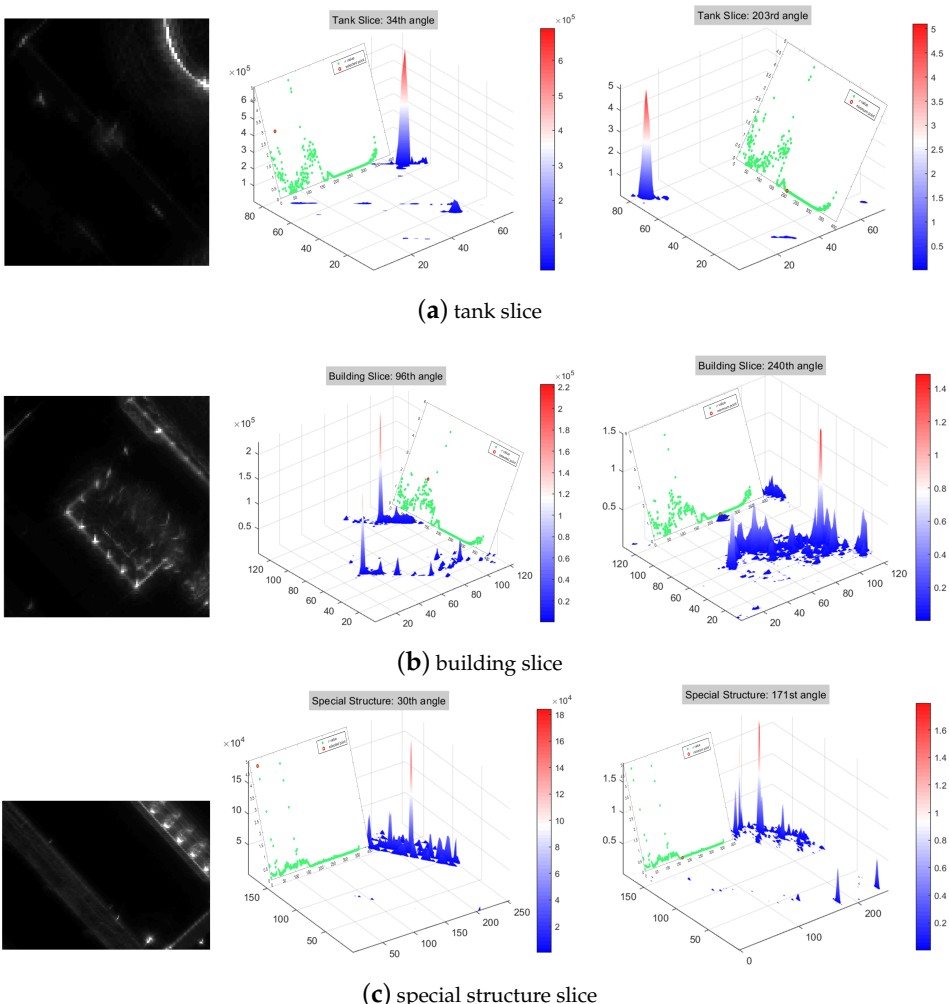

**Figure 15.** Parameter analysis of tank, building, special structure SAR slice: (**a**) tank slice image, 34th angle $\sigma$ value and energy distribution, 203rd angle minimum $\sigma$ value and energy distribution; (**b**) building slice image, 96th angle $\sigma$ value and energy distribution, 240th angle minimum $\sigma$ value and energy distribution; (**c**) special struture slice image, 30th angle $\sigma$ value and energy distribution, 171st angle minimum $\sigma$ value and energy distribution.

From Figures 14 and 15, the relationship between $\sigma$ value and the target energy can be drawn: in the angle where the $\sigma$ is particularly small, the energy of the target slice is very small. At the same time, if there is no other targets exist in the slice, the energy distribution is still dominated by the target, otherwise it is characterized as other non-target energy. In the angle with a large $\sigma$ value, the energy distribution is dominated by the target scattering, and it is necessary to pay attention to the abnormal value that may correspond to the non-target. From the above, the general law of two-parameter characterization with multi-aspect partial target scattering is obtained: the $\beta$ value represents the difference between the target scattering and the surround scattering, and a large value indicates that the target scattering is not very different from the surrounding scattering or even the target cannot be seen. Secondly, the $\sigma$ value characterizes the energy distribution of the target. A large value indicates that the target has strong backscattering and can compare the scattering energy between different angles. At the same time, in the process of quantitative analysis, the parameter curve can be used to judge the basic scattering characteristics of the target, especially in the analysis of the whole target scattering.

## 5. Discussion

In parametric scattering feature extraction, there may also be situations where the values do not match the energy values, such as Slice2 in Table 5. The characterization of different target scattering may also be related to other factors: the stability of the radar platform, the multi-aspect imaging algorithm, the relative position of the target, etc. For the selection of the target type, the target with typical structure in the data is selected for separate analysis, including the overall target and the partial target. In the selection of partial targets, the target is divided into four for separate solving, and the number of slice types can be increased in further research. In order to summarize the parameteric model, the parameter ranges of the studied targets are given at the end and the error value range is added to the parameter values. The analysis of target scattering within parameter value range is reliable, and the scattering difference of targets can be analyzed. Figure 16 shows the value range of two parameters representing four kinds of target scattering, in which Target1-Target4 represent warehouse, tank, building, and special structure, respectively. An error value is introduced based on the solved value (minimum error is 0.2). From the $\beta$ histogram, it can be seen that benchmark value of warehouse is above 5, the value of Slice1 and Slice3 in tank is large, and that of remaining two targets is around 2. For $\sigma$ histogram, the value of Slice1 in warehouse is large about 15, and that of Slice2 and Slice4 in tank is larger than other slices. The remaining two targets are relatively stable and concentrated around 3 and 5, respectively. The above parameter values can be used to characterize the target scattering at the corresponding angle, so as to complete the parameterized feature extraction of target. Parameterized characterization of target scattering proposed in this paper is based on statistical analysis. The two parameters in expectation maximization are applied to multi-aspect target slices. Previous analysis of multi-aspect target scattering was established within the pixel neighborhood range, while the parametric model enables experimentation with target-level scattering. In the following research, the parameterized characterization scattering will be used to target labeling, extraction.

The range of parameter characterization characteristics is given in the experiment. Compared to previous work, the method of quantitative parameter extraction is used to represent the target scattering characteristics. Previous work mainly focused on fitting images to distinguish artificial targets from natural targets. This paper conducts a comparative analysis of targets with different structures, including different structural parts of the same target. In engineering applications, parametric scattering characterization can be applied to target recognition and classification.

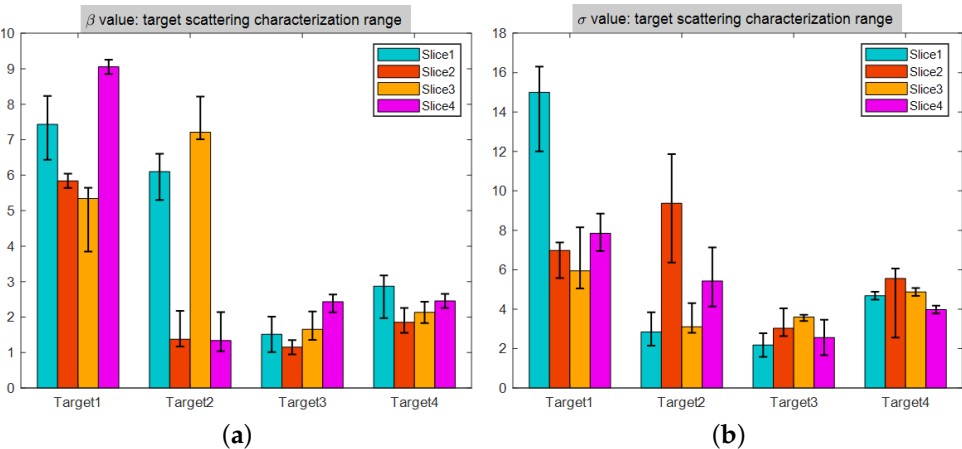

**Figure 16.** The range of $\beta$ and $\sigma$ value for target scattering characterization: (**a**,**b**) $\beta$,$\sigma$ histogram: Four kinds of target with four slices (Target1–Target4 represent warehouse, tank, structure, special structure respectively).

## 6. Conclusions

The multi-aspect target scattering analysis includes two types: the whole target and the partial target. The parametric model established in this paper uses two parameters: $\beta$ and $\sigma$, to characterize the target backscattering on the basis of statistical fitting, which provides a new research idea for the multi-aspect target scattering analysis. At the same time, the shape of the parameter curve can quantitatively characterize the target scattering, and obtain the obvious scattering characteristics of the target at the corresponding angle. For the overall target, the parameters with angle changing are used to distinguish different structural targets. The $\beta$ parameter is combined with the backscattering amplitude difference between the target point and surrounding point to analyze, and the $\sigma$ parameter is analyzed by using the solution values of different targets. For partial targets (the target is divided into four parts), the parameters can be used to analyze the target along the angle and the slice type respectively, where the $\beta$ parameter can characterize the backscattering difference between the target and the surrounding, and the $\sigma$ parameter characterizes the energy distribution of target. Finally the range of two parameter value that characterizes the backscattering of the target is given. With the help of multi-aspect SAR observation, the parameteric model can extract and characterize the backscattering characteristics of the target, and then characterize target's structure types. The multi-aspect target scattering based on the parametric model is suitable for targets with different structures. The next research will expand the parameter sets of different target types and classify targets based on multi-aspect SAR observations.

**Author Contributions:** Conceptualization, X.Y. and F.T.; Methodology, X.Y. and F.T.; Software, X.Y.; Validation, X.Y., F.T., Y.L. and W.H.; Resources, Y.L. and W.H.; Writing—original draft preparation, X.Y.; Writing—review and editing, F.T., Y.L. and W.H.; Supervision, W.H.; Funding acquisition, W.H. All authors have read and agreed to the published version of the manuscript.

**Funding:** This research was funded by National Natural Science Foundation of China grant number 61860206013 and 62171435.

**Data Availability Statement:** Not applicable.

**Conflicts of Interest:** The authors declare no conflict of interest.

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
