# Peer review of "Target Scattering Feature Extraction Based on Parametric Model Using Multi-Aspect SAR Data"

_remotesensing, doi:10.3390/rs15071883_

Round 1
Reviewer 1 Report
In paper, the parameter model is used to extract the scattering characteristics of targets with different structures. The experimental results give the parameter representation with sampling window based on the analysis of the target scattering. The proposed method is interesting and is expected to achieve great application perspectives, and the experimental analysis is sufficient. On this basis, several suggestions are put forward for the author:
1. The content of target feature extraction mentioned in the introduction is less, and it is suggested to add relevant references.
2. It is suggested to add specific description of parameters in the parameter model in the general description of section 2.2. For example, describe the double parameters mentioned in the article.
3. What is the basis of the angle selected in Table 4? Explain why the slice calculation value around the 170th angle is the smallest.
4. Whether the relationship between the SAR image and the energy distribution in Figure 15 is based on parameter extraction or the initial target structure. How much target information is included in the input model?
5. Some statements are redundant, and it is suggested that some expressions be concise:
Conclusion: “At the same time, the parameterized scattering…”
Reviewer 2 Report
General description:
In this paper, an algorithm applying the G0 distribution of target’s scattering field to fit the target area of the SAR scene is developed based on expectation maximization. Different target image types in the scene are resolved by β and σ statistical parameters of the backscattering amplitude model of the target backscattering. The scattering of partial-target slices are evaluated by two energetic parameters (amplitude difference from surrounding points, scattering energy). It is illustrated that the parametric model quantitatively characterizes the scattering feature of the target, and the parameters changing corresponds to the change of the target image feature. C-band circular SAR data is used to validate the developed method.
Conclusion: The authors suggest an original approach for targets’ recognition based on statistical features of the observed targets on the scene.
For editorial and punctuation remarks see an attached file

Reviewer 3 Report
1) Similar analysis work is carried out in the following manuscript.
X. Yue, F. Teng, Y. Lin and W. Hong, "A Man-Made Target Extraction Method Based on Scattering Characteristics Using Multiaspect SAR Data," in IEEE Journal of Selected Topics in Applied Earth Observations and Remote Sensing, vol. 14, pp. 11699-11712, 2021, doi: 10.1109/JSTARS.2021.3127537.
It is strongly recommended that the authors mention the difference or advancement in this present work compared to the aforementioned one.
2) It is suggested to add the range-azimuth resolution of the implemented data also in Table 1 so that the reader of the paper can understand the feasibility of performing one pixel-based analysis as carried out in this manuscript
3) It is wrongly written in lines 172-173 of the paper that "Fig. 4 shows the 360° coherent complex image using BP algorithm". In actual content, Fig. 4 shows the optical image of the observing scene. Again in the following line, Fig. 3 is wrongly mentioned. Please correct the figure numbers.
Round 2
Reviewer 1 Report
It's a pleasure to accept the paper.